# Pre-Existing and Acquired Resistance to PARP Inhibitor-Induced Synthetic Lethality

**DOI:** 10.3390/cancers14235795

**Published:** 2022-11-24

**Authors:** Bac Viet Le, Paulina Podszywałow-Bartnicka, Katarzyna Piwocka, Tomasz Skorski

**Affiliations:** 1Fels Cancer Institute for Personalized Medicine, Department of Cancer and Cellular Biology, Temple University Lewis Katz School of Medicine, Philadelphia, PA 19140, USA; 2Laboratory of Cytometry, Nencki Institute of Experimental Biology, Polish Academy of Sciences, 02-093 Warsaw, Poland

**Keywords:** bone marrow microenvironment, DNA repair, PARP inhibitors, PARPi resistance, leukemia cells, synthetic lethality

## Abstract

**Simple Summary:**

PARP inhibitors (PARPi) have been administered to treat *BRCA1/2*-mutated/deficient malignancies. Nevertheless, the resistance to PARPi is emerging in experimental and clinical interventions. Importantly, the resistance originated from diverse mechanisms, therefore requiring tremendous efforts to identify mechanistic aspects and develop combinational therapies to prevent the resistance and/or restore the efficiency of PARPi in cancer cells. Here, we review pre-existing and acquired resistance to PARPi and propose potential therapeutic solutions.

**Abstract:**

The advanced development of synthetic lethality has opened the doors for specific anti-cancer medications of personalized medicine and efficient therapies against cancers. One of the most popular approaches being investigated is targeting DNA repair pathways as the implementation of the PARP inhibitor (PARPi) into individual or combinational therapeutic schemes. Such treatment has been effectively employed against homologous recombination-defective solid tumors as well as hematopoietic malignancies. However, the resistance to PARPi has been observed in both preclinical research and clinical treatment. Therefore, elucidating the mechanisms responsible for the resistance to PARPi is pivotal for the further success of this intervention. Apart from mechanisms of acquired resistance, the bone marrow microenvironment provides a pre-existing mechanism to induce the inefficiency of PARPi in leukemic cells. Here, we describe the pre-existing and acquired mechanisms of the resistance to PARPi-induced synthetic lethality. We also discuss the potential rationales for developing effective therapies to prevent/repress the PARPi resistance in cancer cells.

## 1. Introduction

Synthetic lethality is a biological process inducing cell death, which is based on the simultaneous inhibition of two pathways that act parallelly in a process required for cell survival. Meanwhile, the inhibition of only one pathway results in cell survival. The synthetic lethality strategy has been widely implemented in anti-cancer therapies. As one pathway may be inactivated in cancer cells due to transformation-related changes, targeting the other pathway triggers cell death while sparing healthy cells [1].

One of the critical features of cancer cells is genomic instability generated by the accumulation of DNA damage, including DNA double-strand breaks (DSBs), which are one of the most lethal DNA lesions in cells [2,3]. However, cancer cells are able to survive and proliferate by modulating their DNA repair pathways, which may differ from those in normal cells [4].

DSBs can be repaired by two major mechanisms: BRCA1/2-mediated homologous recombination (HR) and canonical DNA-PKcs-mediated non-homologous end joining (c-NHEJ) [5]. HR is the major DSB repair mechanism in the S cell cycle phase, whereas c-NHEJ repairs DSBs throughout the cell cycle [6,7]. When HR is inactivated due to deficiencies in BRCA1/2, the prevention and repair of DSBs highly depend on poly-ADP ribose polymerase 1 (PARP1)-mediated base excision repair (BER) and alternative-non-homologous end-joining (a-NHEJ) [8,9]. a-NHEJ is also called microhomology-mediated end-joining (MMEJ) [10], and a-NHEJ/MMEJ involving DNA polymerase theta (Polθ) is called Polθ-mediated end-joining (TMEJ) [11]. Therefore, the inhibition of PARP1 can lead to the induction of the synthetic lethality in proliferating cells harboring HR deficiency (HRD) due to mutations in *BRCA1* and *BRCA2*—for example, [12,13,14,15,16]. Those studies led to the development and implementation of the synthetic lethality triggered by the PARP inhibitor (PARPi), which is currently one of the most effective agents against HR-deficient malignancies [17]. Concomitant c-NHEJ deficiencies enhance PARPi-mediated synthetic lethality in HR-deficient cells [18].

FDA-approved PARPi has been administered to patients with *BRCA1/2*-mutated cancers such as breast and ovarian carcinomas [19,20,21,22]. Although leukemia has not been recognized as a typical *BRCA1/2*-mutated cancer, our group and others have recently reported that certain types of leukemias and other hematopoietic malignancies display HR with/without concomitant c-NHEJ functional deficiency caused by leukemia-inducing mutations [18,23,24,25,26]. In addition, HR and/or c-NHEJ deficiency could be induced by the treatment of leukemia/solid tumors with the tyrosine kinase inhibitors (TKi) against the cancer-driven oncogenic tyrosine kinases (e.g., FLT3(ITD), JAK2(V617F), c-KIT(N822K), IGF-1R, EGFR). Therefore, oncogenic tyrosine kinase (OTK)-driven malignancies can effectively respond to PARPi after the inhibition of OTK [27,28,29,30,31].

Unfortunately, the resistance to PARPi has been reported in both preclinical research and clinical treatment. The acquired (time-dependent) resistant phenotype has been mediated by the functional recovery of HR, the abrogation of PARP1 expression, alterations in DSB end resection, the prevention of the replication fork from degradation, the loss of replication gaps and the arising of alternative factors (e.g., RAD52, Polθ) [29,32,33,34]. Moreover, recent studies have shown that the resistance to PARPi could be pre-existing (time-independent), induced by certain oncogenes triggering malignant transformation and by the tumor microenvironment [35]. In this review, we discuss the pre-existing and acquired mechanisms of PARPi resistance and possible therapeutic solutions.

## 2. Synthetic Lethality in the Context of DNA Repair

Genomic instability is one of the classical hallmarks of cancer [2]. Among other reasons, it occurs when the activation of oncogenes and/or the inactivation of tumor suppressor genes induce the production of endogenous reactive oxygen species (ROS). This leads to the accumulation of oxidative DNA damage in cancer cells, which is accompanied by secondary DNA mutations (e.g., derived from the initial treatment with chemotherapeutic agents) [36]. As a result, mutations in certain genes may lead to the inhibition of processes, which could counteract the cancer cells propagation such as apoptosis, senescence or DNA checkpoint pathways. To survive under the elevated DNA damage rate, cancer cells modulate DNA repair pathways [4]. This not only provides a pro-survival effect but also reveals an “Achilles heel” of cancer cells that can be therapeutically exploited by an anticancer concept called synthetic lethality. In this theory, cell death caused by synthetic lethality is based on mutations in two genes encoding two parallel proteins/pathways that perform functionality in a compensational process required for cell survival. In such case, the functional maintenance of one gene enables a compensatory response to prevent cell death when the other parallel protein/pathway is inactive due to gene mutations [37]. Historically, the first study describing the interplay between two parallel genes was conducted in *Drosophila melanogaster* in 1922 [38], and the term “synthetic lethality” was introduced in 1946 when the same result was found in *Drosophila pseudoobscura* [39]. Since then, the concept of synthetic lethality has been developed in the field of anti-cancer therapy.

In cancer cells, the accumulation of DNA damage can result in a mutation in one gene from the synthetically lethal pair, therefore putting the dependence of cell survival strictly on the second gene from this pair. In this instance, the mutation-free second gene becomes a weakness of cancer cells that should be therapeutically targeted to trigger synthetic lethality. This would eliminate malignant cells without an unwanted cytotoxic effect on normal cells. As DNA repair pathways are altered and/or enhanced to support the survival of cancers cell under high levels of spontaneous and/or drug-induced DNA damage, the synthetic lethality-based targeting of DNA repair pathways is a promising approach to develop a novel therapeutic strategy for anti-cancer treatment.

Among all types of DNA damage, DSBs are one of the most lethal types of DNA lesions in cells. Several exogenous (e.g., γ-irradiation, anti-cancer drugs) and endogenous metabolic factors (e.g., ROS) can cause DSBs as a consequence of direct damage or a stalled replication fork [40]. Once generated, DSBs should be effectively repaired; if not, they will result in cell death [3]. Altogether, redundant DSB repair mechanisms represent a perfect opportunity for the implementation of the concept of synthetic lethality in anti-cancer therapy.

In general, DSB repair consists of two major pathways: HR and NHEJ [5] (Figure 1). HR is considered an accurate DSB repair pathway because it depends on a sister chromatid of a cell in the S/G2 phases of a cell cycle as a template for DNA synthesis and the repair of a DSB [7]. Therefore, HR is capable of repairing DSBs, mostly in proliferating cells [6]. The HR repair pathway is comprehensively modulated by proteins encoded by two classical tumor suppressor genes, *BRCA1* and *BRCA2*, followed by the recruitment of RAD51, its paralogs (including RAD51B, RAD51C, RAD51D, XRCC2 and XRCC3) and RAD54 [6,41,42] to recognize a homologous DNA template and perform the strand invasion to repair DSBs. Mutations in *BRCA1/2* have been found in a wide range of cancers, leading to the inactivation of the HR pathway. In such cases, cancer cells usually employ alternative DNA repair pathway(s) to survive.

On the other hand, NHEJ is an error-prone DSB repair mechanism. NHEJ can be divided into two pathways including the c-NHEJ and a-NHEJ, also known as MMEJ or TMEJ [8,9,10,43]. Both pathways allow for DSB repair in cells throughout all phases of cell cycle [6]. Unlike HR, the DSBs in both quiescent and proliferating cells can be repaired by c-NHEJ, with joint participation of Ku70/80 protein, catalyzing kinase subunit of DNA-dependent protein kinase (DNA-PKcs), NHEJ1 and the complex of XRCC4/LIG4/XLF [44]. 

Although a-NHEJ also performs an error-prone DSB repair, it is more likely to generate more extensive alterations in DNA sequence than c-NHEJ, leading to an increased risk of chromosomal translocations [45]. The a-NHEJ pathway is critically regulated by PARP1, serving a backup role of both HR and c-NHEJ when one/both of the two DSB repair pathways is/are not fully functional [9]. In fact, PARP1 co-operates with MRE11, Polθ, and WRN helicase to promote DSB end re-section process, eventually leading to DNA ligation (which depends on catalytic activities of LIG1 and/or LIG3) [46,47].

In addition, PARP1 has been reported to mediate DNA single-strand break (SSB) repair by recruiting proteins of the base excision repair (BER) pathway [48]. In this pathway, PARP1 conducts a search throughout a single DNA strand until the enzyme recognizes an SSB and binds to the break. At this point, the elevated activity of a process named PARylation (catalyzed by the PARP enzymes family) occurs. This includes covalent bonds of long ADP-ribose to generate poly ADP-ribose (PAR) on PARP1 [8], whereas essential proteins comprising XRCC1 complex, DNA polymerase β and LIG3 are recruited to repair the SSB. Notably, if the SSB remains unrepaired during DNA replication, it will cause transcriptional arrest, leading to the formation of a lethal DSB [49]. Therefore, PARP1-dependent SSB repair is considered very important for the survival of HR-deficient proliferating cells because it prevents the conversion of an SSB to a DSB during the DNA replication. This establishes a rationale for the therapeutic application of PARPi-induced synthetic lethality in *BRCA1/2*-mutated/deficient cancer cells.

## 3. PARPi-Induced Synthetic Lethality in *BRCA1/2*-Mutated Cancers

The usage of PARPi, which predominantly blocks the activity of PARP1, PARP2 and PARP3, is a well-established example of synthetic lethality-based therapy in *BRCA1/2*-mutated cancers with a limited toxicity towards normal cells and tissues [12,13,14,15]. In fact, the effectiveness of PARPi in the *BRCA1/2*-mutated breast/ovarian tumors has initiated an era of personalized medicine with the utilization of PARPi [50,51,52]. Mechanistically, mutations in *BRCA1/2* inactivate the HR pathway, and in order to survive, *BRCA1/2*-mutated cancer cells require the activity of PARP1 in BER and/or a-NHEJ, to prevent the formation of DSBs from unrepaired SSBs during DNA replication. Therefore, the inhibition of PARP1 by PARPi results in stalled replication forks and the accumulation of lethal DSBs, leading to cell death.

Recently, another mechanism has been proposed to regulate PARPi-triggered synthetic lethality in *BRCA1/2*-mutated cells: the single-strand DNA replication gaps [53,54]. Enhanced replication gaps in BRCA1/2-deficient cells were coupled with PARPi sensitivity. Besides working effectively in *BRCA1/2*-mutated cancers, PARPi-mediated synthetic lethality is capable of sensitizing c-NHEJ-deficient cancer cells. For example, the downregulation of LIG4 (involved in the c-NHEJ pathway to perform DNA ligation) induced the sensitivity of melanoma cells to PARPi (olaparib), without a cytotoxic effect on normal melanocytes [55].

Initially, the major mechanism of the efficiency of PARPi has been associated with the interference of the accessibility of NAD^+^ to the PARP1 catalytic domain, leading to the inactivation of the PARylation process and the inhibition of BER and/or a-NHEJ [56]. However, recent studies have shown that the inhibition of the catalytic activity of PARPs is not the only mechanism triggering synthetic lethality [57]. Additionally, PARPi can cause the trapping of PARP1 (and probably also PARP2), resulting in DNA replication, transcriptional arrest and the accumulation of DSBs. The magnitude of synthetic lethality triggered by PARPi corresponds to their capability of PARP1 entrapment [58]. PARPi talazoparib (also known as BMN673) has been reported to be approximately 20–200 times more efficient than previous versions of PARPi, such as olaparib [59]. The elevated efficacy of talazoparib results from its enhanced PARP1-trapping capacity, thus making talazoparib one of the best PARP-trapping agents among currently available PARPi [60].

## 4. PARPi in Clinical Trials of *BRCA1/2*-Mutated Cancers

Olaparib (commercial name—Lynparza^®^) is the first pharmacological PARPi that has been administered in clinical trials. Until now, olaparib is the most common PARPi used in BRCA1/2-deficient cancers. Historically, olaparib became the first PARPi approved by the FDA in December 2014, based on its significant efficacy in the treatment of relapsed ovarian cancer individuals with *BRCA1/2* mutations [61]. In August 2017, olaparib obtained the second approval from the FDA as an extensive therapy for patients with recurrent fallopian tube, peritoneal or epithelial ovarian cancer who have achieved partial or complete remission after the systematic standard chemotherapy [62,63]. Additionally, the potential of olaparib in the anticancer therapy has been extended in January 2018, when the FDA licensed the PARPi as a therapeutic strategy for germline *BRCA1/2*-mutated metastatic breast cancer patients who previously received chemotherapy [64]. This marked olaparib as the first FDA-approved compound working effectively in individuals with hereditary breast cancer. Besides the trials in *BRCA1/2*-mutated breast and ovarian cancer, olaparib was also granted approval by the FDA in different solid tumors. This includes *BRCA1/2*-mutated metastatic pancreatic cancer in 2019 [65], fallopian and primary peritoneal carcinoma in a combinational intervention with bevacizumab [66] and HR-deficient metastatic castration-resistant prostate cancer in 2020 [67].

In addition, two other PARP inhibitors, rucaparib and niraparib, which also target polymerase enzymatic activity, have obtained approval for clinical trials. In detail, the FDA accepted the clinical trials of rucaparib for *BRCA1/2*-mutated advanced ovarian carcinomas undergoing multiple chemotherapy treatments in 2016 [68], reoccurring ovarian, fallopian and primary peritoneal carcinoma without *BRCA1/2* mutational status in 2018 [69] and *BRCA1/2*-mutated metastatic castration-resistant prostate cancer in 2020 [70]. Meanwhile, niraparib achieved the approval of the FDA for reoccurring ovarian, fallopian and primary peritoneal carcinoma with complete or partial chemotherapeutic response in 2017 [71], HR-deficient reoccurring ovarian, fallopian and primary peritoneal carcinoma without chemotherapeutic response in 2019 [72] and advanced ovarian carcinomas with complete or partial chemotherapeutic response in 2020 [73].

On the other hand, based on the significant PARP1 trapping capacity, talazoparib has been clinically employed in breast cancer patients with germline mutations of *BRCA1/2* and other types of cancer that contain impaired DNA damage responses [22,74]. For example, phase III clinical trials of talazoparib demonstrated the increased overall survival rate of metastatic breast cancer patients [75], and it has been approved by the FDA since 2018 [76]. Besides talazoparib, another orally available PARPi (veliparib) is currently undergoing clinical trials [77]. It shows the best selectivity against PARP1/2/3 catalysis, though this PARPi exhibits a limited efficacy of PARP1 trapping [78]. This demonstrated that PARPi, which exerts a more potent and selective inhibitory effect on the PARylation process, is also capable of entering clinical trials.

## 5. Therapeutic Potential of PARPi in Hematopoietic Malignancies

Many recent studies, including ours, have shown that even if *BRCA1/2* mutations are rarely detected in leukemias, PARPi-induced synthetic lethality can be effectively exploited in BRCA1/2-deficient hematopoietic malignant cells. Using a comprehensive Gene Expression and Mutation Analysis strategy, we were able to identify acute myeloid leukemias/acute lymphoblastic leukemias (AMLs/ALLs) that displayed HR and/or c-NHEJ deficiency and were also sensitive to PARPi [18]. These DSB repair defects were detected by direct measurements of the expression of HR and c-NHEJ genes by mRNA microarrays, real-time PCR and/or flow cytometry. In addition, genetic alterations inducing hematopoietic malignancies, such as oncogenes driving myeloid and lymphoid malignancies, including *AML1-ETO* (also known as *RUNX1-RUNX1T1*), *BCR-ABL1*, *PML-RARα*, *TCF3-HLF*, *IDH1/2*^mut^ and *IGH-MYC*, and loss-of-function mutations in tumor suppressor genes (e.g., *TET2, WT1*), can lead to the deregulation of HR and/or c-NHEJ activity, thus rendering cells susceptible to a synthetically lethal effect triggered by PARPi [23,24,25,79,80,81,82,83,84,85,86,87,88,89,90,91] (Figure 2A and Table 1). In addition, mutations in the core cohesion complex gene *STAG2* (Stromal Antigen 2) induce DNA damage, stalled replication forks and a high genetic dependency on PARP1 in AML/myelodysplastic syndrome (MDS) cells. Therefore, those cells are sensitive to PARPi talazoparib both in vitro and in vivo; however, the mechanism remains unexplored [92].

Furthermore, we and others described that those malignant hematopoietic cells expressing oncogenic tyrosine kinases (OTK) (e.g., BCR-ABL1, FLT3(ITD), JAK2(V617F)) spontaneously accumulate high levels of oxidative DNA damage and DSBs due to the increase in ROS production [93,94,95]. However, OTK-positive cells were capable of escaping from the cytotoxic effect of DSBs due to enhanced/modulated DSB repair. Remarkably, the inhibition of these OTKs by FDA-approved specific tyrosine kinase inhibitors (TKi) (JAK1/2 inhibitor ruxolitinib, FLT3 inhibitor quizartinib, ABL1 inhibitor imatinib) resulted in acute HR/c-NHEJ deficiency (due to the downregulation of BRCA1, BRCA2, RAD51 and/or LIG4) and the sensitivity to PARPi (Figure 2B) (Table 2). Therefore, the combination of TKi and PARPi was capable of eradicating both proliferating and quiescent malignant hematopoietic stem and progenitor cells [18,27,28,29,79]. All these promising results have made up a rationale for clinical trials with PARPi in patients with leukemias and other related hematopoietic malignancies [26].

As a result, the first trial (NCT01399840) of PARPi in hematopoietic malignancies began in 2014, when the efficacy of talazoparib was tested in 25 AML/MDS patients and 8 other individuals with chronic lymphocytic leukemia (CLL) and mantle cell lymphoma [96]. During 2017, there were three phase I clinical trials registered, including a combinational therapy of veliparib + temozolomide in 48 patients with relapsed/refractory AML (NCT01139970) [97], veliparib combination with topotecan and carboplatin in a clinical study of 99 patients with relapsed/refractory AML, chronic myelomonocytic leukemia or aggressive myeloproliferative neoplasms (NCT00588991) [98], and olaparib in 15 patients with relapsed CLL, T-prolymphocytic leukemia or mantle cell lymphoma [99]. In 2021, the results of a clinical trial (NCT04326023) examining the efficiency of PARP inhibitors, including olaparib, rucaparib, niraparib, talazoparib and veliparib, in 178 patients with MDS and AML were reported [100]. Although 104 in 178 MDS/AML participants were recorded with positive outcomes, PARPi increased the risk of MDS/AML in adults over 18 [100]. Additionally, a phase I clinical trial of the DNA methyltransferase inhibitor decitabine and talazoparib has been demonstrated in 25 patients with relapsed/refractory AML [101].

## 6. Acquired Resistance to PARPi-Mediated Synthetic Lethality

Despite the strong potency of PARPi in DSB repair-defective cancer cells, the resistance to PARPi-induced synthetic lethality has been reported in BRCA1/2-deficient and HRD tumor cells (Table 3). The most common mechanism responsible for the development of resistance to PARPi is the restoration of HR repair activity in *BRCA1/2*-mutated cancer cells [102]. In detail, the secondary mutations in *BRCA1/2* were associated with the abrogation of the chain terminator/frameshift resulting from original mutations, leading to the restoration of the full-length *BRCA1/2* open reading frame. Therefore, the active BRCA1/BRCA2 protein expression is recovered, thus restarting the fully functional HR pathway [34,103,104]. Besides the resistance mediated by the additional mutations in *BRCA1/2*, the reduction/loss of methylation of *BRCA1* promoter has been suggested, which restores the expression of BRCA1, thereby leading to resistance to PARPi. Indeed, PARPi-sensitive primary breast cancer cells exhibited elevated *BRCA1* promoter methylation, which was associated with impaired BRCA1 expression. Meanwhile, decreased promoter methylation and proficient BRCA1 expression were observed in individuals who did not respond to PARPi treatment [105]. Moreover, in a subset of BRCA1-deficient triple-negative breast cancer cells showing refractory against PARPi, a reduced expression of EMI1 impairs EMI-dependent RAD51 degradation, restoring HR repair activity [106]. Additionally, EMSY is a negative regulator of type I interferon response and also counteracts the HR repair pathway, and KEAP1 targets EMSY for ubiquitin-mediated degradation. Therefore, the overexpression of KEAP1 destabilizes EMSY, leading to HR restoration and resistance to PARPi. On the contrary, the inactivation of KEAP1 due to gene mutations causes non-small cell lung cancer to become sensitive to PARPi thanks to the stabilization of EMSY and HR deficiency [107].

Another mechanism of PARPi resistance in BRCA1/2-deficient tumor cells is due to the downregulation of PARP1 expression, thereby resulting in PARylation-independent cell proliferation and/or the inefficiency of PARP1 trapping by PARPi. In fact, the expression level of PARP1 was remarkably reduced in colorectal carcinoma HCT116 clones that were refractory against PARPi and temozolomide [109]. Moreover, in patients with ovarian cancer, a missense mutation (1771C>T) in *PARP1* was described to induce the de novo attenuation of sensitivity to PARPi [110]. Additionally, the overexpression of P-glycoprotein efflux pumps has been shown to result in resistance to PARPi olaparib in a long-term treatment in murine triple-negative mammary carcinomas [15].

A third mechanism related to the acquired PARPi resistance is associated with the alteration of DSB end resection. In this circumstance, via a complex of 53BP1-RIF1, the 53BP1-mediated suppression of DSB end resection leads to enhanced c-NHEJ activity, reducing the dependency on PARP1-dependent a-NHEJ and thus causing PARPi resistance [111,112]. Hence, the inhibition of 53BP1 in *BRCA1*-null murine embryonic stem cells promotes DSB end resection to restrict c-NHEJ, thus sensitizing cells for PARPi [132]. On the other hand, the activity of 53BP1 is also required to maintain the efficiency of PARPi olaparib in *BRCA1*-mutated breast cancer cells [115]. In these cells, the somatic loss of 53BP1 leads to olaparib resistance due to the partial restoration of HR repair activity, decreasing the response of BRCA1-deficient mouse mammary tumors to the inhibitor [113,114]. Additionally, the blockage of 53BP1 localization to DSBs also contributes to the inactivation of PARPi in BRCA1-deficient tumors, mediated by the increased expression of TIRR [116]. Moreover, the inactivation of SHLD1/2/3 complex and the decreased expression of other proteins in DSB end resection, including RIF1 and REV7, restore the functional activity of HR, diminishing the efficacy of PARPi [117,118,119]. In the same mechanistic manner, TRIP13 is a negative regulator of the SHLD1/2/3 complex by dissociating REV7-SHLD1/2/3 to promote HR [120,121]. Therefore, the amplification of TRIP13 induces resistance to PARPi in *BRCA1/2*-mutated cancers [122]. Furthermore, another DSB end resection-associated mechanism is acquired by the decrease in DNLL1, increasing the DSB end resection potential and recovering HR functionality in *BRCA1*-mutated cells [123]. In addition, it has been recently shown that the CHAMP1-POGZ heterochromatin complex counteracts the 53BP1 inhibitory effect against HR and binds directly to REV7 to repress the REV7-SHLD1/2/3 complex, elevating HR repair activity via promoting DSB end-resection [124,125]. Hence, the overexpression of CHAMP1 confers PARPi resistance; meanwhile, the depletion of CHAMP1 restores functional 53BP1 and the complex of REV7-SHLD1/2/3 to restrict DSB end resection, leading to PARPi sensitivity.

Besides functioning in the HR pathway, BRCA1 and BRCA2 have been reported to play a role in maintenance of the integrity of replication forks [126,133]. The acquired resistance to PARPi in BRCA1- and BRCA2-deficient cells can be caused by the decreased expression of the MRE11 and MUS81 nucleases via the loss of EZH2, leading to a protective response toward replication forks [126,127]. Furthermore, the deficient activities of PTIP and SLFN11 maintain the stability of replication forks by the prevention of fork degradation, decreasing the efficiency of PARPi [128,129].

In addition, going beyond mechanisms of acquired PARPi resistance based on the restoration of HR or replication fork protection, a recent study has reported that the resistance can be induced by a reduction/loss of DNA replication gaps via the restoration of Okazaki fragment processing (OFP) [53]. Basically, replication gap has been reported as a major determinant of PARPi response in BRCA1/2-deficient cells [53]. In general, deficiency in BRCA1/2 extends replication gaps, leading to the PARPi sensitivity of the cells. However, in a cell model with a simultaneous knock-out of *BRCA1* and *53BP1*, the OFP was recovered by the restoration of XRCC1-LIG3, leading to the decrease/depletion of replication gaps and, eventually, to the resistance to PARPi. On the contrary, a double knock-out of *53BP1* and *LIG3* re-sensitizes *BRCA1*-mutated cells to PARPi [53].

Additionally, the arising of alternative factors in DSB repair pathways is an important mechanism of acquired resistance to PARPi. For example, RAD52-mediated DNA repair can play a role as a backup of HR when BRCA1/2-RAD51-RAD54 are not fully functional [134]. At a DSB, RAD52 stimulates DNA pairing (a pivotal step of D-loop formation) and single-strand DNA annealing to regions of homology (>30 bps) [135]. The annealed DNA is then processed by nucleases (e.g., ERCC1/XPF) to generate an error-prone repair [136,137]. Therefore, the overexpression of RAD52 can result in the inefficiency of PARPi in HRD cells. Our group has successfully demonstrated that the simultaneous targeting of PARP1 and RAD52 induces “dual synthetic lethality” in BRCA1/2-deficient cancer cells [33].

Besides RAD52, another alternative factor in DSB repair which is widely being studied is Polθ, encoded by *POLQ*. Polθ facilitates MMEJ, a branch of a-NHEJ [11], and Polθ inactivation causes synthetic lethality in HRD cells [138,139,140]. The over-activation of Polθ also confers resistance to radiation and chemotherapies including PARPi [138,140,141,142]. Therefore, the pharmacological inhibition of Polθ is being developed to overcome the resistance to PARPi in a cohort of HRD cancers [130,131].

## 7. Pre-Existing Resistance to PARPi: The Bone Marrow Microenvironment

Hematopoietic stem cells (HSCs) originate from a specific hematopoietic tissue: the bone marrow (BM). Anatomically, the bone marrow (BM) is established by various types of stromal cells, including mesenchymal stromal/stem cells and endothelial cells, together with sympathetic nerve-related cells, macrophages, osteoblasts, fibroblasts, megakaryocytes and others, all functioning under the hypoxic conditions. The term “HSC niche” was introduced in 1978 by Ray Schofield to initially elicit the importance of hematopoietic tissues, such as BM and the spleen, for HSCs’ biology [143]. This has led to numerous studies being conducted during the past four decades to validate the function of the bone marrow microenvironment (BMM) [144]. Overall, the BMM plays a significant role in the maintenance, self-renewal and differentiation of HSCs, while molecular interactions between HSCs and cellular components of the BMM are set to maintain the balance between the self-renewal and differentiation of HSCs [144].

While the BMM plays a key role in hematopoiesis, there is a reciprocal relationship between leukemia stem cells (LSCs) and the BMM. LSCs are responsible for the initiation of the disease, as well as the reprogramming of the BMM. Meanwhile, the BMM has been documented to provide essential factors for the maintenance and survival of LSCs [145,146]. Therefore, the protective feature of the BMM towards LSCs often renders malignant cells refractory against chemotherapeutic agents and TKi [147,148,149]. Moreover, in primary xenograft conditions, the CD34^+^ leukemia cells remodeled murine BM including bone marrow stromal cells, to promote an unfavorable microenvironment for normal HSCs but provide a protective and pro-survival milieu for LSCs [150,151,152]. Altogether, BMM can be a potential weakness of LSCs that can be therapeutically exploited to eliminate hematopoietic malignant cells in the BMM.

Recently, we discovered a pre-existing mechanism of resistance to PARPi in hematopoietic malignant cells residing in the BMM [35]. HR and/or c-NHEJ-deficient hematological malignancies, which are sensitive to PARPi-mediated synthetic lethality in conditions mimicking the peripheral blood microenvironment (PBM), become resistant to PARPi in conditions mimicking the BMM. We show that transforming growth factor beta 1 (TGF-β1) produced by bone marrow stromal cells activates a hypoxia-induced TGF-β receptor (TGFβR) kinase–SMAD2/3 canonical pathway in leukemic cells to promote DSBs repair (Figure 3). The TGFβR kinase inhibitor did not alter the sensitivity of leukemia cells to PARPi in the PBM, indicating that the regulation of the TGFβR-SMAD3 signaling in DSB repair activities is exclusive in the BMM. Mechanistically, TGFβR kinase signaling stimulates the repair of DSBs in leukemia cells in the BMM via the upregulation of BRCA1/2, ATM, DNA-PKcs and LIG4. This finding suggests potential clinical applications of TGFβR kinase inhibitors in PARPi-mediated interventions against hematopoietic malignancies. In fact, the inhibition of TGFβR kinase by galunisertib, which has been applied in clinical trials of several cancers [153,154,155,156,157,158], restored the PARPi sensitivity of leukemia cells in the BMM. Therefore, we postulate that the addition of galunisertib to PARPi treatment should improve the therapeutic outcome.

## 8. Pre-Existing Resistance to PARPi: Tumor-Inducing Mutations

Our group has shown that OTK c-KIT (N822K mutant in c-*KIT* receptor tyrosine kinase) induces PARPi resistance in BRCA1/2-deficient AML1-ETO-positive AML [18,29]. The inhibition of the oncogenic c-KIT kinase by TKi avapritinib re-sensitized leukemia cells to PARPi. Furthermore, somatic variants in *DNMT3A* are identified as mutations in hematological malignancies that affect the epigenetic regulation of DNA methylation. They often co-occur with activating mutations in OTKs such as *FLT3^ITD^*, *BCR-ABL1*, *JAK2^V617F^* and *MPL^W515L^*. We reported that DNMT3A-deficient cells favor HR/c-NHEJ, owing to the downregulation of PARP1 and the reduction of a-NHEJ [85]. In addition, a recent study has reported that *DNMT3A*-mutated leukemia cells exhibit impaired PARP1 recruitment, p53 activation and increased DNA damage after being challenged by replication stress-induced medications [108]. Consequently, DNMT3A-deficient leukemia cells are resistant to PARPi. However, the disruption of TET2 dioxygenase activity and/or the TET2–Wilms tumor 1 (WT1) binding ability are responsible for HR/c-NHEJ repair defects, the restoration of a-NHEJ activity and the sensitivity to PARPi. Therefore, TET2 dioxygenase inhibitors should be explored therapeutically to reverse PARPi resistance in DNMT3A-deficient leukemias. Besides the recent findings of our group, other researchers also discovered that the activation of tyrosine kinase activities of EGFR and IGF-1R stimulates the activity of HR by increasing BRCA1 and RAD51, respectively, causing resistance to PARPi [30,31]. Therefore, using the tyrosine kinase inhibition of EGFR and IGF-1R induces the “BRCAness” and HRD phenotypes in breast and ovarian cancers, leading to PARPi sensitivity in the cancer cells.

## 9. Conclusions

In light of the ever-increasing list of mechanistic pathways responsible for resistance to PARPi, the pharmacological inhibition of pre-existing and acquired PARPi resistance has to be achieved in order to improve therapeutic efficiency of PARPi-mediated synthetic lethality. For example, pre-existing PARPi resistance in leukemia cells could be addressed by combining inhibitors of PARP and TGFβR kinase (to re-sensitize leukemia cells in the BMM [35]), inhibitors of PARP and TET2 dioxygenase (to re-sensitize leukemias carrying *DNMT3A* mutations [85]) and inhibitors of PARP and OTK (FLT3(ITD), JAK2(V617F), c-KIT(N822K), EGFR, IGF-1R [27,28,29,30,31]). Acquired PARPi resistance could be prevented by the more aggressive “dual synthetic lethality” approach (simultaneously targeting PARP1 and RAD52 [33]), which eliminates more tumors cells in a shorter time, thus reducing the chances of the time-dependent emergence of resistant clones. Moreover, if PARPi resistance emerges, these clones could be eliminated by the inhibition of another DNA repair mechanism, e.g., Polθ-mediated TMEJ [130,131,159].

## Figures and Tables

**Figure 1 cancers-14-05795-f001:**
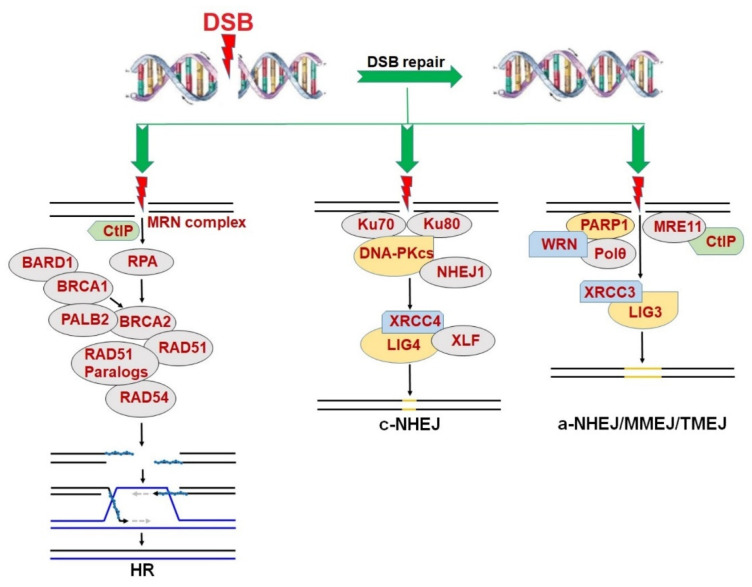
DNA double strand break (DSB) repair pathways including homologous recombination (HR), canonical non-homologous end joining (c-NHEJ) and alternative non-homologous end joining (a-NHEJ)/microhomology-mediated end joining (MMEJ)/DNA polymerase theta (Polθ)-mediated end joining (TMEJ).

**Figure 2 cancers-14-05795-f002:**
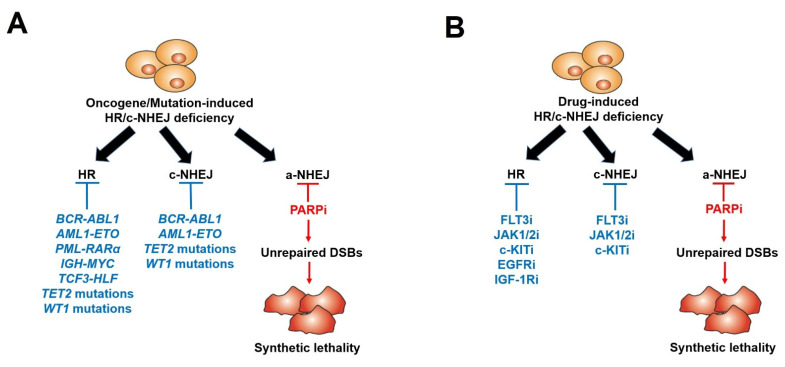
Scheme of PARP inhibitors administered in hematopoietic malignancies and other tumors displaying HR/c-NHEJ deficiency induced by oncogenes/mutations (**A**) and tyrosine kinase inhibitors (**B**).

**Figure 3 cancers-14-05795-f003:**
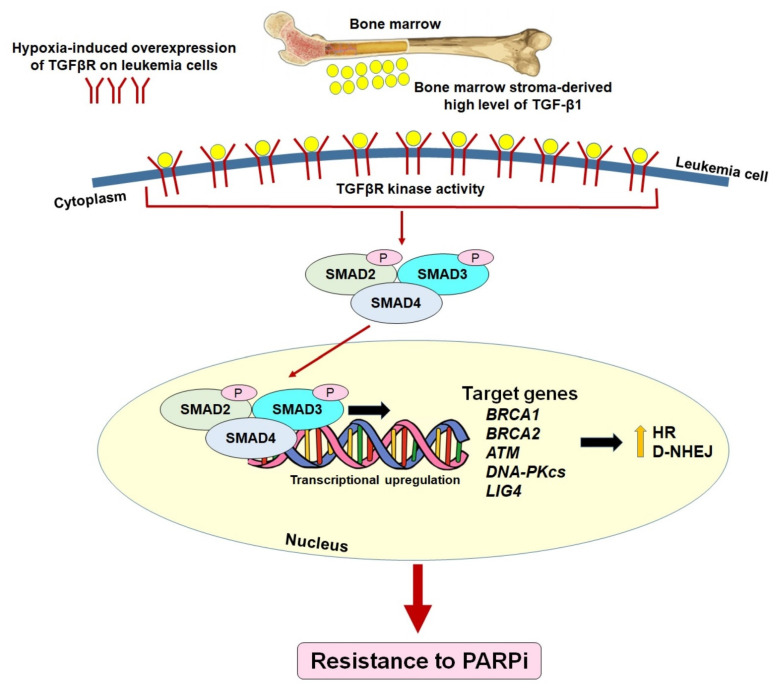
Pre-existing mechanism of PARP inhibitor resistance by the restoration of HR/c-NHEJ in HR/c-NHEJ-deficient leukemias via the activation of TGF-β1—TGFβR—SMAD2/3 signaling in the hypoxic bone marrow microenvironment.

**Table 1 cancers-14-05795-t001:** Oncogenes/Mutations inducing HR/c-NHEJ deficiency.

Disease	Oncogene/Mutation-Induced HR/c-NHEJ Deficiency	Deregulated Protein	References
CML	*BCR-ABL1*	BRCA1, DNA-PKcs	[18,23,82,86,91]
AML	*AML1-ETO*	BRCA1, BRCA2, Ku70	[25]
AML	*PML-RARα*	BRCA2, RAD51C	[25,87]
Burkitt lymphoma	*IGH-MYC*	BRCA2	[24]
AML	*IDH1/IDH2* mutants	ATM	[84,88,89]
AML/ALL	*TCF3-HLF*	BRCA1, BRCA2	[80]
AML	*FLT3^ITD^*+*TET2* mutant	BRCA1, LIG4	[85]
AML	*FLT3^ITD^*+*WT1* mutant	BRCA1, LIG4	[85]
AML/MDS	*TET2* mutant	BRCA1	[90]

**Table 2 cancers-14-05795-t002:** Therapeutic drugs inducing HR/c-NHEJ deficiency.

Disease	Drug-Induced HR/c-NHEJ Deficiency	Deregulated Protein	References
Myeloproliferative neoplams [JAK2(V617F)]	JAK1/2 kinase inhibitor (Ruxolitinib)	BRCA1, RAD51C, LIG4	[27]
AML [FLT3(ITD)]	FLT3 kinase inhibitor (Quizartinib)	BRCA1, BRCA2, PALB2, RAD51, LIG4	[28]
AML [c-KIT(N822K)]	c-KIT kinase inhibitor (Avapritinib)	BRCA1, BRCA2, DNA-PKcs	[29]
Breast cancer	EGFR kinase inhibitor (Lapatinib)	BRCA1	[30]
Breast and ovarian cancers	IGF-1R kinase inhibitor	RAD51	[31]

**Table 3 cancers-14-05795-t003:** Mechanisms of pre-existing and acquired resistance to PARPi.

Pre-Existing PARPi Resistance	Mechanism	References
TGFβ1—TGFβR—SMAD2/3 signaling in a hypoxic bone marrow microenvironment	Restoration of HR/c-NHEJ	[35]
Loss-of-function mutations in *DNMT3A*	Enhanced HR/c-NHEJ Deregulation of PARP1	[85,108]
Activation of FLT3 kinase [FLT3(ITD)]	Enhanced HR/c-NHEJ	[28]
Activation of JAK2 kinase [JAK2(V617F)]	Enhanced HR/c-NHEJ	[27]
Activation of c-KIT kinase [c-KIT(N822K)]	Enhanced HR/c-NHEJ	[29]
Activation of EGFR kinase	Increased BRCA1	[30]
Activation of IGF-1R kinase	Increased RAD51	[31]
**Acquired PARPi Resistance**	**Mechanism**	**References**
Secondary mutations in *BRCA1/2*	Restoration of HR	[34,102,103,104]
Reduced BRCA1 promoter methylation	Restoration of HR	[105]
Deregulation of EMI1	Restoration of HR	[106]
Overexpression of KEAP1, decrease in EMSY	Restoration of HR	[107]
Upregulation of P-glycoprotein efflux pumps	Enhanced drug efflux	[15]
Decrease in PARP1	Loss of PARP1 expression	[109]
Mutations in *PARP1*	Loss of PARP1 expression	[110]
Elevated activity of 53BP1-RIF1 complex	Enhanced c-NHEJ	[111,112]
Loss of 53BP1	Enhanced DSB end resection Restoration of HR	[113,114,115]
Increase in TIRR	Blockage of 53BP1 localization to DSBs	[116]
Decrease in RIF1, REV7 and REV7-SHLD1/2/3 complex	Enhanced DSB end resection Restoration of HR	[117,118,119]
Overexpression of TRIP13	Dissociation of REV7-SHLD1/2/3 complex	[120,121,122]
Decrease in DNLL1	Enhanced DSB end resection Restoration of HR	[123]
Overexpression of CHAMP1-POGZ complex	Decreased 53BP1 and REV7-SHLD1/2/3 complex	[124,125]
Decrease in MRE1, MUS81 and EZH2	Replication fork protection	[126,127]
Decrease in PTIP and SLFN11	Prevention of fork degradation	[128,129]
Increase in XRCC1-LIG3	Restoration of OFP Decrease/loss of replication gaps	[53]
Overexpression of RAD52	Alternative factor: HR backup	[33]
Overexpression of Polθ	Alternative factor: MMEJ/TMEJ	[130,131]

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
