# Peer review of "Pre-Existing and Acquired Resistance to PARP Inhibitor-Induced Synthetic Lethality"

_cancers, 2022, doi:10.3390/cancers14235795_

Round 1

Reviewer 1 Report

Viet Le et al review the mechanisms of PARP inhibitor resistance, both cancer cell intrinsic (primary and acquired) and extrinsic (niche-dependent), with an emphasis of hematologic malignancies. The review is well-written and timely; it will be a valuable addition to the translational cancer research field.

Given the growing enthusiasm for PARP inhibition as a potential treatment for blood malignancies, I suggest adding published studies by Tothova et al. JCI Insight, 2021 and Venugopal et al. CCR, 2022 (Tables 1 and 3). To balance out this enthusiasm, it would be important to also highlight the emerging data on the limitations of PARPi therapy such increased risk of tMN (Morice et al. Lancet Hematol 2021). Although the latter study is referenced in the manuscript, it appears to be out of context.

Author Response

Dear Sir/Madam,

Thank you very much for your time and effort to review our manuscript. All your comments are really appreciated. The revised manuscript has been attached and our response(s) as follows: 

Viet Le et al review the mechanisms of PARP inhibitor resistance, both cancer cell intrinsic (primary and acquired) and extrinsic (niche-dependent), with an emphasis of hematologic malignancies. The review is well-written and timely; it will be a valuable addition to the translational cancer research field.

Given the growing enthusiasm for PARP inhibition as a potential treatment for blood malignancies, I suggest adding published studies by Tothova et al. JCI Insight, 2021 and Venugopal et al. CCR, 2022 (Tables 1 and 3). To balance out this enthusiasm, it would be important to also highlight the emerging data on the limitations of PARPi therapy such increased risk of tMN (Morice et al. Lancet Hematol 2021). Although the latter study is referenced in the manuscript, it appears to be out of context.

RESPONSE: As requested the suggested references are included and discussed in the revised manuscript. However, since mechanism of PARPi sensitivity in Tothova et al. JCI Insight, 2021 was not explored, the study cannot be listed in Table 1, but they were discussed on page 6, at the end of paragraph 4. Meanwhile, the citation of Venugopal et al. CCR, 2022 [111] has been added in Table 3.  In addition, the limitations of PARPi therapy are now mentioned on page 8, paragraph 1.

Reviewer 2 Report

In this manuscript the authors review the current knowledge about the mechanisms of action of and resistance to PARP inhibitors in both clinical and preclinical settings. The review is reasonably comprehensive, with a unique emphasis on a less well-known area, namely the study of PARP inhibitors in the hematopoietic compartment and blood cancers. In my view, the following issues need to be addressed before the manuscript can be considered for publication.

First, the authors used a new term “constitutive resistance” to describe the observation that BRCA-deficient leukemia cells are resistant to PARPi when in the bone marrow. In my view, this term is not only inaccurate but also misleading. Perhaps it should be called conditional resistance, niche-specific resistance, or microenvironment-dependent resistance, etc.

Second, for most readers, the hematopoietic section may feel excessive and tiring to read. It can be much more concise with clearer points. In this section, a bit more discussion of BRCA-related studies should be added, while the rest can be shrunk substantially.

Third, given the fact that loss of PARP1 itself causes PARPi resistance, it is now widely accepted that the inhibition of enzymatic activity of PARP is of minimal importance for PARPi sensitivity of BRCA/HR deficient cells (while PARP trapping and possibly replication gap formation are now considered to be the major determinants). The authors repeatedly mentioned the regulation of A-NHEJ by PARP, which may not even be relevant to PARPi sensitivity or resistance in most situations. The review of literature could be more critical.

Fourth, given the evolution of our understanding of the mechanisms of action of PARPi, I would suggest that the authors modify the relevant passages. For example, the authors could make clear that PARPi inhibits PARP function in SSB repair leading to replication fork collapse and DSB formation in the S phase was the initial thinking, however, it was later found that …… Just a suggestion.

Fifth, RAD52 and Pol-theta were only briefly mentioned in the conclusions. It would be much better if they were also mentioned and discussed earlier.  

Sixth, the use of terms such as HRness and D-NHEJ are very peculiar. HRness is in fact inappropriate, and the correct term should be HRD (HR deficiency). D-NHEJ should be replaced by the widely used c-NHEJ for either conventional or canonical NHEJ. Coining new terms for these is not only unnecessary but also leads to confusion.

Finally, there are a number of grammatical errors that need to be cleaned up. For example, “trapping on PARP1” in page 6 third paragraph, “capable to escape” in page 8 first paragraph, “on contrary” in page 9 first paragraph, and “On the other hand, deficient activities of PTIP and SLFN11 remain the stability of replication forks” in page 10 first paragraph, etc. etc.

Minor – page 9 second paragraph, AZD2281 is in fact olaparib and should be written as such.

Author Response

Dear Sir/Madam,

Thank you very much for your time and effort to review our manuscript. All your comments are really appreciated. The revised manuscript has been attached and our response(s) as follows:

Point 1: In this manuscript the authors review the current knowledge about the mechanisms of action of and resistance to PARP inhibitors in both clinical and preclinical settings. The review is reasonably comprehensive, with a unique emphasis on a less well-known area, namely the study of PARP inhibitors in the hematopoietic compartment and blood cancers. In my view, the following issues need to be addressed before the manuscript can be considered for publication.

First, the authors used a new term “constitutive resistance” to describe the observation that BRCA-deficient leukemia cells are resistant to PARPi when in the bone marrow. In my view, this term is not only inaccurate but also misleading. Perhaps it should be called conditional resistance, niche-specific resistance, or microenvironment-dependent resistance, etc.

RESPONSE 1: We replaced “constitutive” with “pre-existing” which reflects PARPi resistance present before the treatment starts. Conditional resistance might suggest “inducible” which could be misled with acquired. In addition, pre-existing is not only caused by BMM, but also by specific oncogenic mutations causing leukemias. Thus, a cohort of leukemias display “pre-existing” PARPi resistance associated with leukemia induction. Altogether, “pre-existing” means PARPi resistance was present before the treatment; conversely, “acquired” means PARPi resistance developed during/in response to treatment.

Point 2: Second, for most readers, the hematopoietic section may feel excessive and tiring to read. It can be much more concise with clearer points. In this section, a bit more discussion of BRCA-related studies should be added, while the rest can be shrunk substantially.

RESPONSE 2: Hematopoietic section has been substantially reduced.

Point 3: Third, given the fact that loss of PARP1 itself causes PARPi resistance, it is now widely accepted that the inhibition of enzymatic activity of PARP is of minimal importance for PARPi sensitivity of BRCA/HR deficient cells (while PARP trapping and possibly replication gap formation are now considered to be the major determinants). The authors repeatedly mentioned the regulation of A-NHEJ by PARP, which may not even be relevant to PARPi sensitivity or resistance in most situations. The review of literature could be more critical.

RESPONSE 3: The role of replication gaps in PARPi response is now included and discussed on page 5, paragraph 3.

Point 4: Fourth, given the evolution of our understanding of the mechanisms of action of PARPi, I would suggest that the authors modify the relevant passages. For example, the authors could make clear that PARPi inhibits PARP function in SSB repair leading to replication fork collapse and DSB formation in the S phase was the initial thinking, however, it was later found that …… Just a suggestion.

RESPONSE 4: According to the suggestion, we add the additional information about the role of replication gaps in regulating sensitivity to PARPi on page 5.

Point 5: Fifth, RAD52 and Pol-theta were only briefly mentioned in the conclusions. It would be much better if they were also mentioned and discussed earlier.  

RESPONSE 5: The role of RAD52 and Pol-theta is now discussed on pages 10-11.

Point 6: Sixth, the use of terms such as HRness and D-NHEJ are very peculiar. HRness is in fact inappropriate, and the correct term should be HRD (HR deficiency). D-NHEJ should be replaced by the widely used c-NHEJ for either conventional or canonical NHEJ. Coining new terms for these is not only unnecessary but also leads to confusion.

RESPONSE 6: Corrected as suggested.

Point 7: Finally, there are a number of grammatical errors that need to be cleaned up. For example, “trapping on PARP1” in page 6 third paragraph, “capable to escape” in page 8 first paragraph, “on contrary” in page 9 first paragraph, and “On the other hand, deficient activities of PTIP and SLFN11 remain the stability of replication forks” in page 10 first paragraph, etc. etc.

Minor – page 9 second paragraph, AZD2281 is in fact olaparib and should be written as such.

RESPONSE 7: Corrected as requested and other grammatical/spelling corrections have been highlighted.

Reviewer 3 Report

This is a well written review and helps address a very important and current concept.

The entire structure of the review is divided into different sections and the introduction is too long and includes information that is also mentioned in other sections (tables)

To maintain cohesion between each part, it would be important to reduce the introduction and include the information from the tables in the section where they are mentioned.

Author Response

Dear Sir/Madam,

Thank you very much for your time and effort to review our manuscript. All your comments are really appreciated. The revised manuscript has been attached and our response(s) as follows:

This is a well written review and helps address a very important and current concept.

Point 1: The entire structure of the review is divided into different sections and the introduction is too long and includes information that is also mentioned in other sections (tables)

RESPONSE 1: All three Tables have been removed from Introduction as suggested.

Point 2: To maintain cohesion between each part, it would be important to reduce the introduction and include the information from the tables in the section where they are mentioned.

RESPONSE 2: The Introduction has been reduced and three Tables have been relocated into appropriate sections as suggested.

Round 2

Reviewer 2 Report

The manuscript has been improved. However, there is one issue that remains to be clarified. In the last, version, it appeared that by "constitutional resistance" the authors meant that BRCA1/2 deficient leukemia cells are PARPi resistant in the bone marrow; however, in the revised manuscript, the concept now appears to refer to a constitutive PARPi resistance of BRCA wt leukemia cells. If this is the case, there is no point of even talking about resistance, because BRCA wt cells are supposed to be resistant to PARPi unless there are other defects in DNA repair or replication fork protection. This needs to be clarified.

Author Response

Dear Sir/Madam,

Thank you very much for your additional comment on our revised manuscript. The comment is really valuable and appreciated. The revised manuscript has been attached and our response as follows:

The manuscript has been improved. However, there is one issue that remains to be clarified. In the last, version, it appeared that by "constitutional resistance" the authors meant that BRCA1/2 deficient leukemia cells are PARPi resistant in the bone marrow; however, in the revised manuscript, the concept now appears to refer to a constitutive PARPi resistance of BRCA wt leukemia cells. If this is the case, there is no point of even talking about resistance, because BRCA wt cells are supposed to be resistant to PARPi unless there are other defects in DNA repair or replication fork protection. This needs to be clarified.

RESPONSE: To clarify the concern, we added one sentence in Chapter 7, 3rd paragraph. In fact, the sentence was written in the original manuscript. However, it was deleted when we reduced the hematopoietic section in the 1st revision.

Round 3

Reviewer 2 Report

Revision acceptable.